# Tree Species Composition in Mixed Plantations Influences Plant Growth, Intrinsic Water Use Efficiency and Soil Carbon Stock

Francesco Niccoli [1], Tiziana Danise [1], Michele Innangi [1], Francesco Pelleri [2], Maria Chiara Manetti [2], Giovanni Mastrolonardo [3], Giacomo Certini [3], Antonietta Fioretto [1] and Giovanna Battipaglia [1,*]

1   Department of Environmental, Biological, Pharmaceutical Sciences and Technologies, University of Campania "Luigi Vanvitelli", Via Vivaldi 43, 81100 Caserta, Italy; francesco.niccoli@unicampania.it (F.N.); tiziana.danise@unicampania.it (T.D.); michele.innangi@unicampania.it (M.I.); Antonietta.FIORETTO@unicampania.it (A.F.)

2   Consiglio per la Ricerca in Agricoltura e l'Analisi dell'Economia Agraria, Centro di Ricerca Foreste e Legno, Viale Santa Margherita 80, 52100 Arezzo, Italy; francesco.pelleri@crea.gov.it (F.P.); mariachiara.manetti@crea.gov.it (M.C.M.)

3   Dipartimento di Scienze e Tecnologie Agrarie, Alimentari, Ambientali e Forestali, Università degli Studi di Firenze, Piazzale delle Cascine 18, 50144 Firenze, Italy; giovanni.mastrolonardo@unifi.it (G.M.); giacomo.certini@unifi.it (G.C.)

*   Correspondence: Giovanna.BATTIPAGLIA@unicampania.it

**Abstract:** Species interactions in mixed plantations can influence tree growth, resources capture and soil fertility of the stands. A combined approach of tree-ring analyses and carbon stable isotope was used to check tree growth and water use efficiency of two species, *Populus alba* L. and *Juglans regia* L., intercropped with each other and with N-fixing or competitive production species. Furthermore, soil analyses were performed to understand how the different intercropping systems can influence soil characteristics, in particular soil carbon stock. Dendrochronological data showed that during the first years, the growth of principal species was favored by intercropping. This positive effect decreased in the following years in most of intercropped stands, due to light competition with the crown of companion species. Carbon isotope data showed that *P. alba* and *J. regia* had the highest intrinsic water use efficiency when growing with *Elaeagnus umbellata* Thunb, a shrubby species with a shallow root system that favors a non-competitive exploitation of soil water resources. Finally, the intercropping of the principal species with *Corylus avellana* L. promoted the highest soil C stock. Our findings confirmed the importance to consider the plantation dynamics and wood formation in the long-run and to apply appropriate thinning and pruning interventions to counteract interspecific competition.

**Keywords:** isotopic analysis; tree-ring; N-fixing species; environmental-growth relationship; intercropping

## 1. Introduction

The wide spreading of secondary forests worldwide is connected to the huge depopulation of the countryside and mountain marginal lands in favor of big cities and to the abandonment of the agricultural and pastoral activities [1]. This land use change has certainly caused a series of positive effects, such as contributing to mitigate global warming through a higher absorption and storage capacity of atmospheric carbon [2]. There are, however, also some downside effects such as increase of fire frequency and the closure of open areas with the homogenization of the landscape [3]. To mitigate these negative effects and to safeguard the economic and ecological functions of forests, it is necessary to implement sustainable management of secondary forests [4].

In the last decades huge afforestation plans have been financially supported to enhance marginal agricultural areas. Compared to the past, planted forests have globally increased due to the growing demand for timber, pulp, energy and other goods [5,6]. In particular, in the European Union, under the support of European Commission Regulation 2080/92,

approximately 1,000,000 ha were afforested between 1992 and 1999 [7]. Most planted forests were monocultures [8–11], the main advantages of which being the ease of implementation and management and the possibility of orienting production towards high valued woody assortments. On the other hand, the main disadvantages of monocultures are the high sensitivity to pest and disease attacks, a progressive reduction of soil fertility, the excessively simple plantation structure and the poor biodiversity which lead to low profits in the timber market [12–14].

Mixed plantations are increasing globally and various studies have demonstrated their superiority in providing various ecosystem services such as greater biodiversity, climate change mitigation, protection from hydrogeological instability, as well as, in some cases, an increasing in tree productivity [15–17]. Indeed, mixed-tree plantations in comparison to monocultures could promoted a reduction in management costs [18], an excellent adaptability of plants to extreme climatic events [19,20], a better resistance to pests and pathogens outbreaks [8,21], improved nutrients cycle and soil fertility [22,23]. Several studies have highlighted the positive effects on soil fertility of intercropping with N-fixing species [24–26]. This type of mixed plantation implies increase in soil organic matter (SOM) [27–30]. A meta-analysis study demonstrated statistically significant increase in the concentrations of C and N in soil of mixed plantations enriched with a N-fixing species [31]. However, the success of mixed plantations is variable [32]. As stated by the stress gradient theory [33] and confirmed by subsequent studies [34], the positive effects determined by competitive production and facilitation species cannot prevail over negative effects (e.g., competition) in stands where stressful abiotic conditions, such as drought, occur. In this context, the study of intrinsic water use efficiency ($_i$WUE), defined as the relationship between the net assimilation of C in photosynthesis (A) and stomatal conductance (gs) [35], allows to evaluate how water competition affects tree growth [8,36] as well as the positive influence of intercropped species on the photosynthetic activity and nitrogen supply of the principal species [37–40]. Currently, information on how to match different species to increase the chances of success of mixed plantations and maximize site conditions is scarce [16,41–43]. Therefore, to maximize the chances of success of mixed plantations it is necessary to deepen knowledge on their dynamics and the ecophysiological processes that are established among the different tree species, as well as to understand the effects of intercropping on biomass productivity and soil fertility [25].

Our research aimed to evaluate the wood formation and tree productivity, trees intrinsic water use efficiency and impact on soil fertility of an experimental tree plantation varied in composition. We examined distinct stands in which two valuable species widely used for the production of timber, white poplar (*Populus alba* L., Salicaceae) and common walnut (*Juglans regia* L., Juglandaceae), are intercropped with each other and with a shade tolerant shrub species such as hazel (*Corylus avellana* L., Betulaceae) and with N-fixing species such as Italian alder (*Alnus cordata* (Loisel.) Duby, Betulaceae) and autumn-olive (*Elaeagnus umbellate* (Thunb.), Elaeagnaceae). We hypothesize that species mixtures, especially N-fixing, could positively affect wood growth and water use efficiency of the studied tree species. Further, the intercropped species could positively affect soil N fertility and increase C stock in soil.

## 2. Materials and Methods

### 2.1. Study Area

The experimental area is located nearby Brusciana, 40 km east of Florence, Central Italy (43°40′31″ N, 10°55′22″ E). The area has a Mediterranean climate, with mild and rainy winters and hot and dry summers. The average annual temperature measured in the period between 2008 and 2019 at a climatic station located 5 km from our study site was 15.4 °C, while the average annual rainfall was 878 mm (Servizio Idrologico Regione Toscana (SIR), http://www.sir.toscana.it/consistenza-rete, access on 15 March 2020). The soils developed on recent (Holocene) fluvial deposits and are *Fluventic Haplustepts coarse-loamy, mixed, thermic* of the USDA Soil Taxonomy, according to the 1:250,000 soil map by

Regione Toscana (http://sit.lamma.rete.toscana.it/websuoli, access on 15 March 2020). The study site was a mixed tree plantation (7.44 ha) established in 1996 on formerly cultivated land. The tree plantation comprises different and neighboring stands/mixed intercropping where poplar and walnut were planted together using a triangular layout with a distance of 8 m (179 trees per ha), and intercropped with different nurse trees and shrubs, using a rectangular layout of 3.5 × 4 m (715 trees per ha) and maintaining the same layout and density of the two main tree species (Figure S1). The individuals of the same species were all the same genotypes.

In this study, the experimental design is the same used by a previous study [26]. Four types of intercropped systems were examined and compared using a randomized blocks design with three replications:

- Poplar and walnut (plot PJ).
- Poplar, walnut intercropped with hazel (plot PJC).
- Poplar, walnut intercropped with autumn-olive (plot PJE).
- Poplar, walnut intercropped with Italian alder (plot PJA).

### 2.2. Sampling and Dendrochronological Processing

In December 2018, 10 dominant trees of *P. alba* (mean diameter at breast height 43.5 ± 6.2 cm, mean height 22.4 ± 2.6 m) and 10 dominant trees of *J. regia* (mean diameter at breast height 17.5 ± 2.6 cm and mean heigh 12.7 ± 1.8 m) were randomly selected in each replicate of each mixture. For each tree, two wood-cores were collected by an incremental borer (Haglöfs Langsele, Sweden). The cores were taken at a height of 130 cm from the ground and in east-west direction. The collected cores were fixed on specific wooden supports and subjected to a sanding process, to facilitate tree-rings identification. Subsequently, dendrochronological measurements were made to the nearest 10 μm using a semiautomatic device (Lintab$^{TM}$, Rinntech, Heidelberg, Germany). Through this system, it was possible to measure the width of the tree-rings and elaborate the representative curves of the growth trend of each individual plant (tree-ring width—TRW). In a second step, the obtained TRW series were visually and statistically crossed with each other through the "Gleichaeufigkeit" (GLK), a specific parameter that assesses the correlation between the different series [44]. Synchronization was considered acceptable with a GLK > 0.70. Finally, to understand the effect of the different intercropping on the diametrical increase, the mean tree-ring width chronology of each replicate and the annual Basal Area Increment (BAI) have been calculated for each intercropping systems and species, *J. regia* and *P. alba*. BAI was calculated from raw ring width, considering concentrically distributed tree-rings [45]. The use of BAI avoids detrending procedure [46] allowing to not lose information on low frequency variability. Then average of Cumulative Basal Area (CBA) was determined by summing the average BAI and propagating uncertainties [47,48]. To distinguish the measurements of the two studied tree species in each intercropping systems, hereafter we refer to intercropping and then P (for poplar) and the J (for walnut) after an underscore, e.g., PJ_P refers to *P. alba* sampled in the PJ intercropping and PJ_J refers to *J. regia* sampled in the PJ intercropping.

### 2.3. Soil Sampling and Laboratory Analysis

In winter of 2018, eight soil cores (10 cm thick) were sampled by a steel cylinder of known volume (so as to determine the bulk density, BD) in each replication of the four intercropping systems, following a regular net approximately centered in the centroid of the stand, from the surface and after litter removal. Once in the laboratory, soil moisture was measured with standard gravimetric method, oven-drying the soil samples (60 °C) to constant weight, and then passing them through a 2 mm sieve to remove rock fragments and large roots. The fine earth was analyzed for particle size distribution (by the hydrometer method [49]), pH (potentiometrically, in 0.01 M CaCl$_2$, using a liquid to soil ratio of 2.5:1), total C (TC) and N (TN) (by dry combustion at a C/N/S Carlo Erba NA1500 Analyzer). Organic C was measured by pre-treating the samples with 6 M HCl at 80 °C to eliminate

carbonates [50]. Total inorganic carbon (TIC) was the difference between total carbon (TC) and total organic carbon (TOC).

For estimating the recalcitrant organic carbon—which is the fraction of TOC assumed to be more stable in soil—the soil was submitted to a chemical oxidation aimed at removing the more labile soil organic matter and isolate an older, chemically resistant fraction of it [51]. Briefly, 10 g of soil were mixed with 20 mL of 1 M HCl for removing the inorganic carbon. The residue was then washed with 40 mL of distilled water and then added with 100 mL of 1 M NaOCl adjusted to pH 8 and put on a mechanical shaker for six hours at 25 °C. Finally, it was centrifuged at $2000 \times g$ for 15 min and the solution discarded. This treatment was repeated four times. Thereafter the sample was washed four times with 50 mL distilled water and oven-dried at 60 °C to constant weight. The C content of the residue was measured by dry combustion and accounted for the mass balance, i.e., subtracting the mass of C in the residue from the mass of TOC in the untreated sample.

### 2.4. Intrinsic Water Use Efficiency Determination

From each of the four-intercropping systems, five tree cores that presented the best cross-dating (GLK > 0.80) were selected for carbon isotopic analyses. Tree rings were very narrow, preventing annual resolution, thus groups of three-years rings were separated using a blade cutter. Then the three-year samples were ground into a powder using a centrifugal mill (ZM 1000, Retsch, Germany), using a mesh size of 0.5 mm to ensure homogeneity.

The C isotope composition was measured at the Icona Laboratory (DISTABIF, Caserta, Italy) by continuous-flow isotope ratio mass spectrometry (Delta V Advantage, Thermo electron corporation, Bremen Germany), using 0.06 mg of dry matter for $\delta^{13}C$ measurements. The standard deviation for the repeated analysis of an internal standard was less than 0.1% for $\delta^{13}C$ measurements, whereas the standard deviation for Sigma-Aldrich $\alpha$-cellulose (item C8002) were less than 0.2‰ for $\delta^{13}C$.

The results from the isotope ratio deviations are presented using the common $\delta$ notation:

$$\delta = \left( \frac{Rs}{Rr} - 1 \right) \times 1000\text{‰} \tag{1}$$

where $R$ refers to the ratio of the $^{13}C$ to $^{12}C$ isotopes in the sample ('$s$') and the reference ('$r$'), compared to the PDB (Pee Dee Belemnite) standard.

Therefore, tree $\Delta^{13}C$ was calculated as follows:

$$\Delta^{13}C = \frac{\delta 13c_a - \delta 13c_p}{1 + \delta 13c_p} \tag{2}$$

where $\delta^{13}c_a$ and $\delta^{13}c_p$ are the ratios in atmospheric $CO_2$ and tree-ring, respectively.

The relative rates of carbon fixation and stomatal conductance are the primary factors that determine $\Delta$ [52]:

$$\Delta^{13}C = a + (b - a)\frac{c_i}{c_a} \tag{3}$$

where $a$ is the discrimination against $^{13}CO_2$ during $CO_2$ diffusion through the stomata ($a$ = 4.4‰), $b$ is the discrimination associated with carboxylation ($b$ = 27‰), and $c_i$ and $c_a$ are the intercellular and ambient $CO_2$ concentrations, respectively.

The intrinsic water-use efficiency ($_i$WUE) can be defined as the ratio of the fluxes of net photosynthesis and conductance for water vapor, which indicates the cost of assimilation per unit of water. Using the simplified, linear relationship [35], the intrinsic water use efficiency is calculated as:

$$_i\text{WUE} = \frac{A}{gs} = \frac{c_a - c_i}{1.6} = c_a \frac{1 - \frac{c_i}{c_a}}{1.6} \tag{4}$$

where *A* is the rate of $CO_2$ assimilation, *gs* is the rate of leaf stomatal conductance, and 1.6 is the ratio of diffusivities of water and $CO_2$ in the atmosphere. Equation (4) is the "basic" form of isotopic discrimination that does not include effects due to mesophyll conductance and photorespiration (i.e., assuming an infinite internal conductance), which were not available for the species here.

$\delta^{13}ca$ can be estimated for the period 1995–2004 according to a reference study [53] and the measured values for the period 2005–2018 available online (http://www.esrl.noaa.gov/gmd/, access on 15 July 2020), while *ca* is the concentration of $CO_2$ in the atmosphere, estimated for each year and obtained by NOAA (http://www.esrl.noaa.gov/, Mauna Loa station, access date 15 July 2020).

### 2.5. Climate Analysis

To evaluate how climatic factors affected the productivity of the studied tree species, the climatic information recorded in the study area in the period between 1995 and 2018, available from the CRU TS3.23 gridded dataset at 0.5° resolution data [54], were examined (Figure 1). In particular, we correlated minimum, maximum, and average monthly temperatures and monthly and total rainfall with CBA data and water use efficiency of the two species, *J. regia* and *P. alba*. Climate data, after being grouped by months, were correlated using the software Microsoft XLSTAT, with Spearman correlation function ($p < 0.05$).

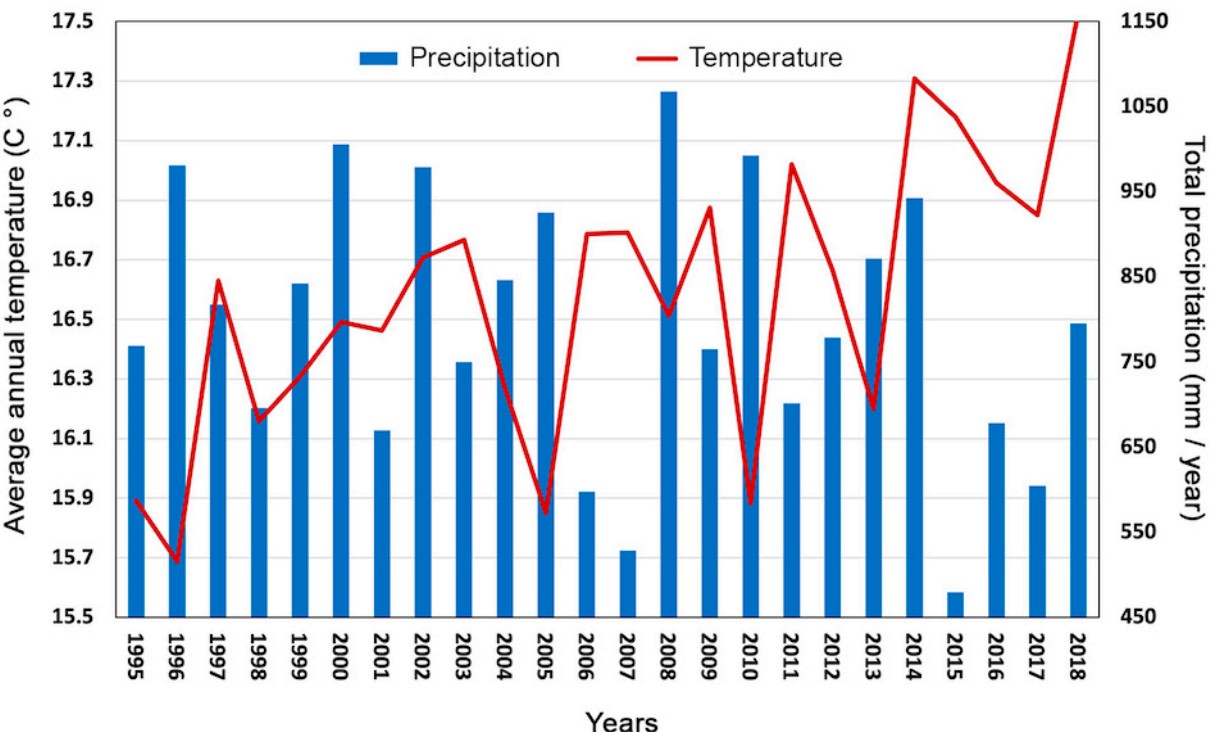

**Figure 1.** Trend of total annual precipitation in blue (expressed in millimeters/year) and average temperatures in red (expressed in °C) recorded from 1995 to 2018 and coming from the KNMI Climate Explorer database.

### 2.6. Data Analyses

All data were checked for normality (using the Shapiro–Wilk test) and homogeneous variance (by means of *Levene's test*) before applying inferential statistics. Differences between intercropping systems in terms of soil parameters, tree growth and iWUE, were evaluated according to one-way ANOVA, using Student–Newman–Keuls coefficient for comparison tests ($p < 0.05$). Cumulative basal area data were compared with one way ANOVA, during the whole period and comparing the juvenile phase (8 years) and the adult one. To explore multivariate patterns between the intercropping systems, a principal

component analysis (PCA) [55] was applied using the package XLSTAT. Data matrices considering the study period and eight variables (iWUE, CBA, TN, TOC, TIC, C/N, recalcitrant C and standing tree density) of each intercropping system were processed as ordination method for indirect gradient analysis [56]. This analysis allows to reduce the number of independent variables by eliminating those accounting for only a few percent of the total variance [57].

## 3. Results

### 3.1. Dendrochronological Analysis of Populus Alba

The dendrochronological analyses carried out on *P. alba* showed that the growth curves in all intercropping systems had a similar trend: in 2008, 2010, 2012 and 2014 there was a simultaneous decrease in growth, while in 2009, 2011, 2013 and 2015 an increase in growth was observed in all the mixtures (Figure 2). The GLK was very high in all intercropping systems: 75, 78, 82, and 75 in PJ_P, PJC_P, PJA_P, and PJE_P, respectively. In some cases, the wood-cores did not have the first rings because it was not possible to sample the pith, which explains why the curves had different starting points. Anyhow, all *P. alba* trees showed a sustained and regular growth trend in the first years. Subsequently, from 2010 onwards, a substantial reduction in growth occurred. In addition, the collected data showed different growth rates of *P. alba* according to the type of intercropping: intercropping with a third species, such as hazel (PJC_P), Italian alder (PJA_P), or autumn-olive (PJE_P), resulted in higher productivity compared to the simple association with *J. regia* (PJ_P).

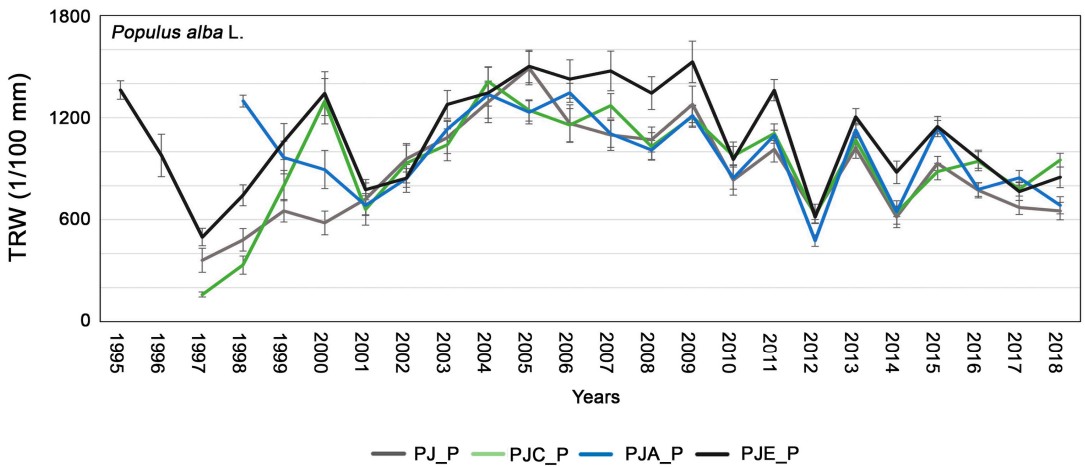

**Figure 2.** Average tree-ring chronologies of *Populus alba* in the different systems: PJ_P in gray; PJC_P in green; PJA_P in blue; PJE_P in black. The bars indicate the standard error.

Looking at the mean cumulative basal area (CBA, Figure 3, referring to the whole ANOVA model testing the overall effect), the greatest mean growth of *P. alba* was found in the PJE_P system, i.e., when it was intercropped with the N-fixing *E. umbellata* (*p* = 0.03). On the contrary, there were no statistical differences (*p* > 0.05) in the growth trends in the case of the intercropping with *A. cordata* (PJA_P) and *C. avellana* (PJC_P). Finally, *P. alba* showed the lowest growth when intercropped with *J. regia* alone (PJ_P, *p* = 0.02).

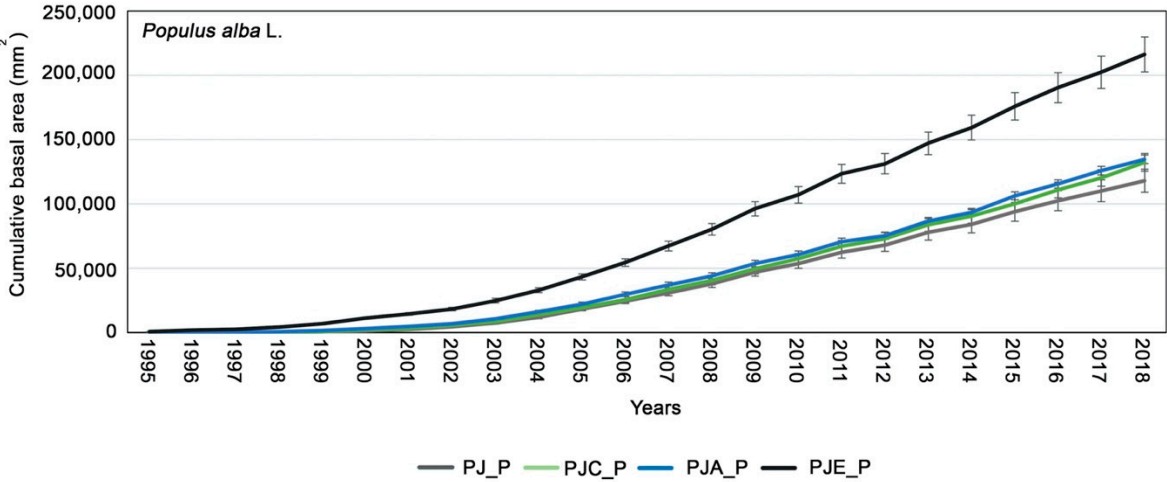

**Figure 3.** Cumulative basal area (CBA) calculated for *Populus alba* in the different systems: PJ_P in gray; PJC_P in green; PJA_P in blue; PJE_P in black. The bars indicate the standard error. Differences between intercropping systems were evaluated according to one-way ANOVA, using Student–Newman–Keuls coefficient for comparison tests ($p < 0.05$).

### 3.2. Dendrochronological Analysis of Juglans Regia

The study of the tree rings carried out suggested an absolute growth lower for *J. regia* than for *P. alba* (Figure 4, $p < 0.05$). The average GLK measured in the different intercropping systems was high: 76, 77, 77, and 82 in PJ_J, PJC_J, PJA_J, PJE_J, respectively. The mean chronologies of *J. regia* presented in all intercropping systems a reduction in tree growth in 2003, 2005, 2011, 2012, 2015, and 2017 and a homogeneous increase in tree growth in 2000, 2004, 2014 and 2018.

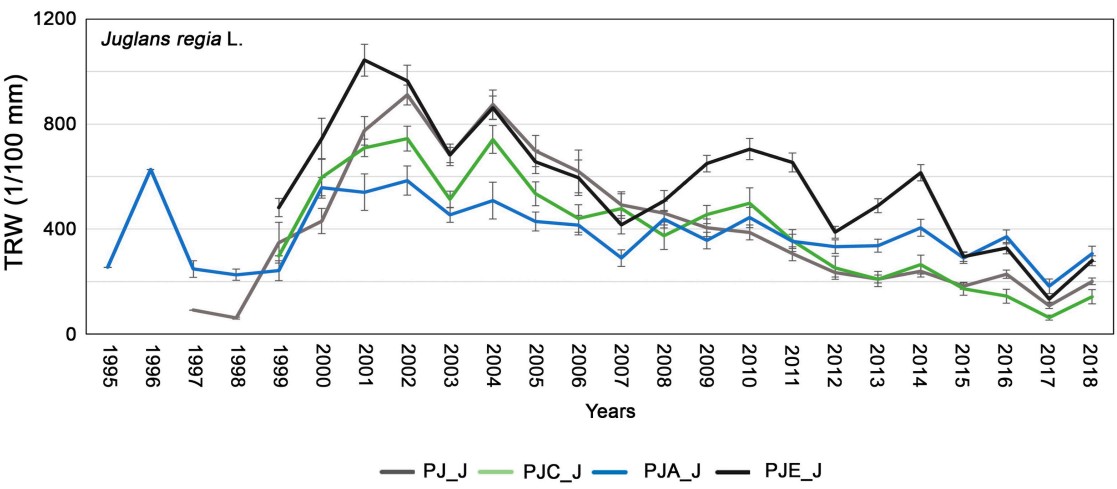

**Figure 4.** Average tree-ring chronologies of *Juglans regia* in the different systems: PJ_J in gray, PJC_J in green, PJA_J in blue, and PJE_J in black. The bars indicate the standard error.

The mean cumulative basal area showed a similar mean growth trend for walnut in the juvenile phase and then a substantial increase in tree growth in the PJE_J (Figure 5, $p = 0.02$). Furthermore, the cumulative curves showed that the plants from PJA_J and PJ_J had low and similar productivity, while that of PJC_J showed the lowest growth (Figure 5).

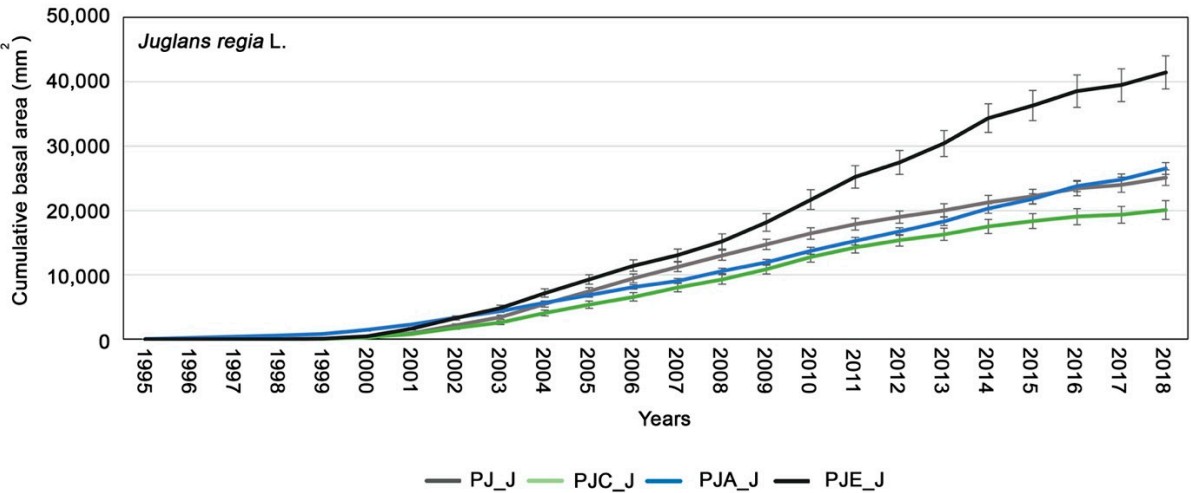

**Figure 5.** Cumulative basal area (CBA) calculated for *Juglans regia* in the different systems: PJ_J in gray, PJC_J in green, PJA_J in blue, and PJE_J in black. The bars indicate the standard error. Differences between intercropping systems were evaluated according to one-way ANOVA, using Student–Newman–Keuls coefficient for comparison tests ($p < 0.05$).

### 3.3. Relationship between Growth, $_i$WUE and Climate

The comparison between the average values of $_i$WUE and mean TRW (Figure 6), calculated for each intercropping systems, showed that both parameters had the highest value when the studied species were intercropped with *E. umbellata* (PJE_J, PJE_P, $p = 0.008$ and $p = 0.006$ respectively).

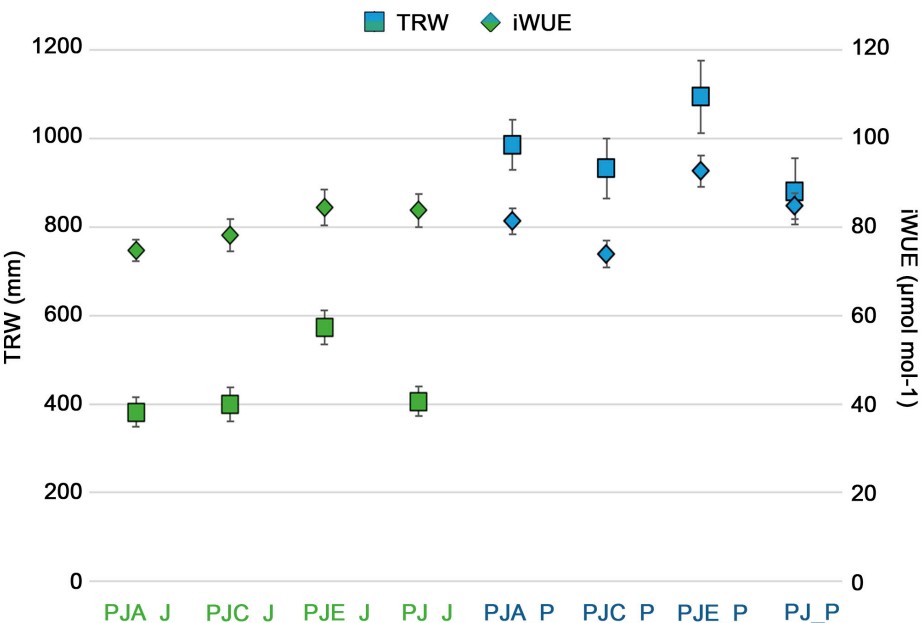

**Figure 6.** Comparison between the average values of TRW (square indicator) and the average values of $_i$WUE (rhombus indicator) calculated for *Juglans regia* (in green) and *Populus alba* (in blue) in the different plots. The bars indicate the standard error. Differences between intercropping systems were evaluated according to one-way ANOVA, using Student–Newman–Keuls coefficient for comparison tests ($p < 0.05$).

Climate analysis, on the other hand, suggested a positive influence of temperature on the growth of *P. alba* (Table 1). In all intercropping systems the CBA values correlated positively ($p = 0.043$) with the maximum and medium temperatures of autumn

(SON = September October and November), as well as with the maximum, minimum and medium temperature of the summer period (JJA = June, July, August). On the contrary, no significant correlations ($p > 0.05$) were observed between growth and precipitation. However, positive correlation (r = 0.95 *) between $_i$WUE and the precipitation in the June-July-August trimester was observed in the PJE_$p$ plot. Finally, negative correlations with the minimum (r = −0.80 *) and medium temperatures (r = −0.82 *) of autumn were highlighted with $_i$WUE in PJ_P (Table 2).

**Table 1.** Correlations between cumulative basal area (CBA) of each intercropping systems with average temperature (Tmean), minimum temperature (Tmin), maximum temperature (Tmax), and total precipitation (Ptot) grouped seasonally (MAM = March, April and May; JJA = June, July and August; SON = September, October and November). Spearman correlation coefficients are reported (r), with the corresponding statistical significance (* 0.05, ** 0.01) *n* = 25.

| | CBA PJ_J | CBA PJC_J | CBA PJA_J | CBA PJE_J | CBA PJ_P | CBA PJC_P | CBA PJA_P | CBA PJE_P |
|---|---|---|---|---|---|---|---|---|
| Tmin MAM | ns | ns | ns | ns | ns | ns | ns | ns |
| Tmin JJA | 0.46 * | 0.42 * | 0.44 * | 0.40 * | 0.51 * | 0.43 * | 0.44 * | 0.45 * |
| Tmin SON | ns | ns | ns | ns | ns | ns | ns | ns |
| Tmax MAM | ns | ns | ns | ns | ns | ns | ns | ns |
| Tmax JJA | 0.45 * | 0.50 * | 0.57 ** | 0.46 * | 0.48 * | 0.51 * | 0.55 * | 0.48 * |
| Tmax SON | 0.44 * | 0.55 * | 0.45 * | 0.50 * | 0.43 * | 0.45 * | 0.48 * | 0.50 * |
| Tmedium MAM | ns | ns | ns | ns | ns | ns | ns | ns |
| Tmedium JJA | 0.43 * | 0.45 * | 0.51 * | 0.47 * | 0.46 * | 0.43 * | 0.52 * | 0.52 * |
| Tmedium SON | 0.50 * | 0.43 * | 0.42 * | 0.45 * | 0.42 * | 0.40 * | 0.43 * | 0.44 * |
| P MAM | ns | ns | ns | ns | ns | ns | ns | ns |
| P JJA | ns | ns | ns | ns | ns | ns | ns | ns |
| P SON | ns | ns | ns | ns | ns | ns | ns | ns |
| P TOT | ns | ns | ns | ns | ns | ns | ns | ns |

**Table 2.** Correlations between water use efficiency (iWUE) of each intercropping systems with average temperature (Tmean), minimum temperature (Tmin), maximum temperature (Tmax), and total precipitation (Ptot) grouped seasonally (MAM = March, April and May; JJA = June, July and August; SON = September, October and November). Spearman correlation coefficients are reported (r), with the corresponding statistical significance (* 0.05), *n* = 8.

| | iWUE PJ_J | iWUE PJC_J | iWUE PJA_J | iWUE PJE_J | iWUE PJ_P | iWUE PJC_P | iWUE PJA_P | iWUE PJE_P |
|---|---|---|---|---|---|---|---|---|
| Tmin MAM | ns | ns | ns | ns | ns | ns | ns | ns |
| Tmin JJA | ns | ns | ns | ns | ns | ns | ns | ns |
| Tmin SON | ns | ns | ns | ns | −0.80 * | ns | ns | ns |
| Tmax MAM | ns | ns | ns | 0.86 * | ns | ns | ns | ns |
| Tmax JJA | ns | ns | ns | ns | ns | ns | ns | ns |
| Tmax SON | ns | ns | ns | ns | ns | ns | ns | ns |
| Tmedium MAM | ns | ns | ns | 0.89 * | ns | ns | ns | ns |
| Tmedium JJA | ns | ns | ns | ns | ns | ns | ns | ns |
| Tmedium SON | ns | ns | ns | ns | −0.82 * | ns | ns | ns |
| P MAM | ns | ns | ns | ns | ns | ns | ns | ns |
| P JJA | ns | ns | ns | 0.84 * | ns | ns | ns | 0.95 * |
| P SON | ns | ns | ns | ns | ns | ns | ns | ns |
| P TOT | ns | ns | ns | ns | ns | ns | ns | ns |

For *J. regia* we observed for all intercropping systems a positive correlation ($p = 0.033$) between CBA values and summer and autumn temperatures (Table 1). There were no significant correlations ($p > 0.05$) between precipitation and CBA. Finally, the correlations between the values of the intrinsic water use efficiency and the climatic data showed positive correlations just in PJE_J with the maximum (r = 0.86 *) and medium temperatures (r = 0.89 *) of the spring period and whit summer rainfall (r = 0.84 *) (Table 2).

### 3.4. Soil Properties

The soil analyses showed significant differences between intercropping systems (Table 3). As regards to soil texture (a variable that is actually independent from the tree cover, but with major repercussions on it), PJA showed the lowest sand and the highest silt contents, contrary to what happened in PJC. These differences were most likely antecedent to the establishment of trees and the used randomized blocks design approach did not completely solve the problem. On the other hand, all the tree systems had a rather homogeneous contribution from the clay fraction, which is the pedological driving factor of SOC accumulation [58,59] and water holding capacity of soil. Actually, no major differences in soil moisture (28% $\pm$ 7) were found between intercropping systems. Soil bulk density and pH were similar, no significant differences being found between plots. The C concentration in soil showed the most remarkable differences. PJC had the highest soil C content (18.19 g C kg$^{-1}$), followed by far by PJA, PJ, and PJE (15.2, 13.4, and 13.2 g C kg$^{-1}$, respectively). No significant difference in soil N was observed between the intercropping systems, despite the presence of N-fixer trees in two of them, i.e., PJA and PJE. The resulting C/N ratio was significantly higher in PJC (15.7), while in all the other plots it ranged between 11.5 and 13. The recalcitrant soil C fraction did not vary much between intercropping systems, ranging from 2.5% and 2.9% of TOC, with the only exception of PJ, where its contribution to TOC was even 8.4%.

**Table 3.** Soil data of the different intercropping systems. Values are mean and standard deviation of 24 replicates, with the exception of sand, silt, clay and recalcitrant C that are mean and standard deviation of three samples. BD = bulk density; TOC = total organic carbon, TIC = Total inorganic Carbon; TN = total nitrogen. The superscript letters ([a,b,c]) indicate significant differences between means in the different column and for each system, according to One-way ANOVA, followed by Student–Newman–Keuls coefficient for comparison tests ($p < 0.05$). Systems that do not share at least one letter are significantly different.

| | Sand | Silt | Clay | BD | pH$_{CaCl2}$ | TOC | TIC | TN | C/N | Stable C |
|---|---|---|---|---|---|---|---|---|---|---|
| | ‰ | ‰ | ‰ | g cm$^3$ | | g kg$^{-1}$ | g kg$^{-1}$ | g kg$^{-1}$ | | % of TOC |
| PJC | 59,555 [a] | 256 $\pm$ 79 [a] | 149 $\pm$ 26 [a] | 1.4 $\pm$ 0.1 [a] | 7.0 $\pm$ 0.2 [a] | 18.19 $\pm$ 0.68 [a] | 18.3 $\pm$ 0.5 [a] | 1.18 $\pm$ 0.43 [a] | 15.7 $\pm$ 2.6 [a] | 2.5 $\pm$ 0.7 [a] |
| PJE | 537 $\pm$ 38 [abc] | 294 $\pm$ 14 [ab] | 169 $\pm$ 32 [a] | 1.5 $\pm$ 0.2 [a] | 7.1 $\pm$ 0.2 [a] | 13.16 $\pm$ 2.90 [b] | 15.9 $\pm$ 0.4 [a] | 1.12 $\pm$ 0.34 [a] | 12.2 $\pm$ 2.2 [b] | 2.9 $\pm$ 1.8 [a] |
| PJA | 375 $\pm$ 82 [b] | 424 $\pm$ 67 [b] | 200 $\pm$ 20 [a] | 1.4 $\pm$ 0.2 [a] | 7.0 $\pm$ 0.2 [a] | 15.19 $\pm$ 4.65 [ab] | 19.1 $\pm$ 0.7 [a] | 1.33 $\pm$ 0.39 [a] | 11.5 $\pm$ 1.8 [b] | 2.5 $\pm$ 1.1 [a] |
| PJ | 437 $\pm$ 53 [c] | 380 $\pm$ 43 [b] | 183 $\pm$ 32 [a] | 1.4 $\pm$ 0.1 [a] | 7.0 $\pm$ 0.1 [a] | 13.41 $\pm$ 4.00 [b] | 21.8 $\pm$ 0.6 [a] | 1.15 $\pm$ 0.45 [a] | 13.0 $\pm$ 4.9 [b] | 8.4 $\pm$ 2.4 [b] |

### 3.5. Multivariate Analysis

The PCA identified the weight of the different variables calculated for the different systems (Figure 7). The first and second components were used for the biplot, so both objects (years) and variables were represented in the same best plane explaining the maximum of variance present in the data. The first two components together explained 60.4% of the variance, with the first component (F1) explaining the higher variance (38.1%). The multivariate analysis showed a clear separation between the different mixtures, with PJ showing a strong positive correlation with recalcitrant C and stand density, and a negative correlation with TIC. Studied tree species in PJC showed a good relationship with TOC and the C/N ratio, while the PJA intercropping was positively related to TN. Finally, the distribution of PJE seemed to be positively influenced by ¡WUE and CBA.

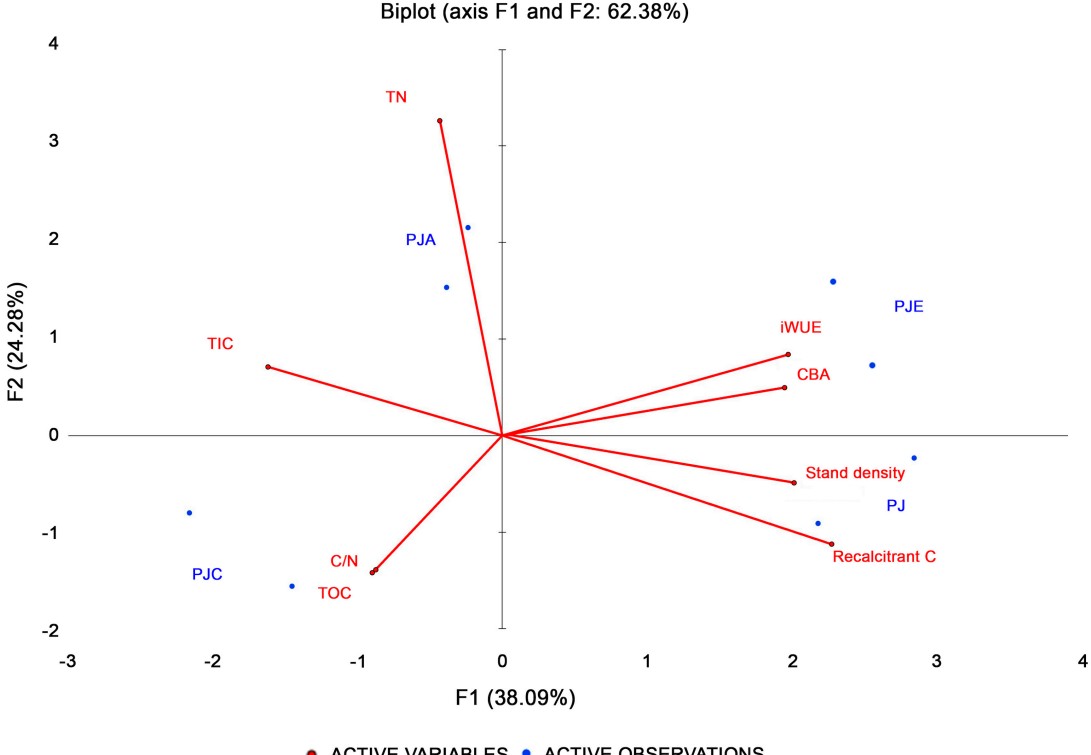

**Figure 7.** Biplot of the first two principal components, which accounted for the total variance of 38.1% and 22.3%, respectively. The length of the vectors (in red) indicates the strength of the correlations with respect to the active observations (intercropping systems, in blue). A strong positive correlation is indicated by vectors pointing in approximately the same direction as the active observations. Conversely, vectors pointing in opposite directions suggest a strong negative correlation.

## 4. Discussion

### 4.1. Influence of the Intercropping on Productivity

Climate–growth correlations proved that temperature was an important factor for the growth of the studied tree species, showing a positive influence in summer and, especially, autumn, when Mediterranean species can reactivate the cambium because the temperature and water are not limiting factors [60,61].

The mean annual and cumulative basal area of the studied tree species in all intercropping systems seemed to be linked to the ecological relationships between those species and the ancillary ones. Multi-species forest crops determine favorable interactions among species through a mechanism known as "principle of competitive production", triggering highest productivity [22,62,63]. This effect is based on niche separation among species, thus interspecific competition can be lower than intraspecific competition for a given environmental factor (such as light, water, nutrient), determining a more efficient use of resources with consequent greater production of biomass of studied tree species [22,64]. Further positive effects can arise from the "facilitative production principle" [22,62,63], which works when the studied tree species are intercropped with facilitation plants; these are plants capable of positively influencing tree growth in terms of height and diameter of the principal species, despite the increase of stand density, through an advantageous alteration of the environmental conditions, such an increase of light availability for the crown [65].

In the PJ system the two pivotal tree species, in particular *P. alba*, showed low productivity despite the low stand density. Here, soil organic matter was relatively richer in the recalcitrant fraction, i.e., the one that best resists decomposition because of its intrinsic chemical composition [66,67], and this may indicate slow recycling of nutrients and their inadequate availability in the vegetative period. However, we cannot speculate further, as

we explored neither the litter decay rate nor the association between SOM and the mineral fraction, which are controlling factors of nutrient availability to plants. Also, the moderate differences in soil texture we found between the various intercropping systems may have somehow affected the different growth rate.

In the PJC stands, the intercropping with *C. avellana* led to a positive effect on the growth of the studied tree species, at least in the first years; indeed, hazel has an excellent ability to rapidly colonize the soil with roots and reduce its competition with weed species [68], exploiting different resources. Furthermore, this ancillary species can influence the structure and the height of the canopy of the principal species, in particular of walnut [69]. However, the positive effect on growth decreased after few years. When the ancillary species become competitive, equaling or exceeding the principal species in height, thinning is necessary in order to favor the canopy development of the valuable species [69]. The intercropping with hazel promoted soil C sequestration. Indeed, it has been demonstrated that hazel positively affects forest floor mass (as is evident in our case), soil nutrients and microbial properties in mixed forests when N is not a limiting factor [70].

The introduction of N-fixing species is generally considered a good forest practice to increase soil fertility and foster the growth of principal species [15,71]. In fact, some studies found higher productivity in principal species when they were intercropped with *Alnus glutinosa* L., *Robinia pseudoacacia* L., or *E. umbellata* [72–74]. In our case, the mixed plantations with a N-fixing species, *A. cordata* and *E. umbellata*, determined a high productivity of the studied tree species with some noticeable differences. In fact, especially *P. alba* seemed to be better influenced by *A. cordata* during the juvenile phase (8 years), while in the longer-term the average tree growth was higher in the PJE stands. Italian alder, thanks to its ability to supply nitrogen, guarantees benefits to the growth and development of the canopy [75] and determines greater availability of light and space for the principal plants with its pyramidal conformation [73]. However, the absence of thinning interventions, before the onset of negative competition phenomena, in the long-term triggered a reduction in tree growth of the studied tree species in PJA compared to PJE. As demonstrated by a previous study [76] on the management of walnut-Italian alder plantations, after a certain period of time thinning of both species have positive effects on the growth of walnuts, stimulating their radial development. The PJE stands presented the highest productivity of the studied tree species, confirming results from several previous studies [73,77–79]. In fact, in addition to being a N-fixing plant, *E. umbellata* has a dense canopy able to reduce competition with weeds [80,81], forcing the valuable plants to take on a more slender shape thus positively influencing their height and diameter [72,78]. In the absence of any management intervention, the PJE intercropping offered greater benefits than PJA, most probably because Italian alder is more competitive than *E. umbellata* for light, water, and nutrients, especially towards the *J. regia* [69].

### 4.2. Influence on the $_i$WUE

As *P. alba* and *J. regia* are very susceptible to water stress [82,83], the analysis of intrinsic water use efficiency may be the key to explore their different growth responses and productivity in the four systems [84]. *Populus alba* showed almost everywhere higher $_i$WUE than *J. regia*. Poplar resorts to various drought resistance strategies that may impact productivity differently: decreased leaf area, leaf abscission, enhanced root growth, increased iWUE, stomatal closure, and osmotic adjustment, among others [83,85,86]. It has been observed that poplar genotypes with high water use efficiency were less susceptible to water stress than genotypes with low $_i$WUE. Thus, poplars with high $_i$WUE could be identified and selected for the purpose of ensuring high biomass productivity [87]. As for walnut, its growth is strongly affected by water deficit, which results in decreased yield [88,89] and in a strong competition for soil water when the rooting zones of neighboring plants overlap [90]. This could account for the lower tree growth and $_i$WUE compared to *P. alba*.

There were no significant differences of $_i$WUE of the studied tree species in the PJ, PJA and PJC stands, while there was in the $_i$WUE of both studied species growing

with *E. umbellata*. The PCA analysis showed that the higher productivity of the studied trees in the PJE stand was linked to an increase in their ¡WUE. Such an increase can be attributed to inherent differential physiological responses of different species [explained following [35] by either an increase assimilation (A) at constant stomatal conductance (gs) or by a reduction in gs at constant A] or by interaction effects of other environmental or climatic factors [91,92]. As the environmental variables were the same in all intercropping systems, we suggest that the increase in ¡WUE and productivity found in PJE could be connected to a major resource availability compared to the other stands. A previous research [40] highlighted an improvement in ¡WUE in the principal species *Quercus robur* growing together with *A. cordata* to be essentially due to the increase in N availability (inferred by the increase in N% in tree-rings) that induces an increase in the photosynthetic activity. Indeed, N fertilization has been frequently reported to increase tree ¡WUE [37–39].

*Alnus cordata* and *E. umbellata* are both N fixing species, but the latter appears to have favored more the studied tree species in terms of ¡WUE and productivity. This could be due to the different strategies for water supply implemented by growing in the mixed stands. To minimize water competition, *J. regia* is able to opportunistically exploit the different available water sources [93], thanks to a very deep and plastic root system [94]. In fact, in the first years of growth, the roots reach a depth of about 3–5 m [95,96], while later they can even deepen up to 10 m [97]. However, intercropping with *A. cordata* can limit roots development in *J. regia* [98]. Therefore, one can hypothesize that in the PJA, a lower root development of the walnut intercropped with an extensive and shallow root system of *P. alba* and *A. cordata* [93], forced the three species to compete for water supply. Such a competition may have resulted in an increase in ¡WUE and lower productivity of the studied tree species in comparison to the PJE system. On the other hand, intercropping with *E. umbellata* (PJE), an ancillary tree with a superficial root system, could have favored a non-competitive exploitation of the soil water resources with the principal plants. The positive correlations found between the summer precipitations and the ¡WUE values of the studied tree species (Table 2) confirmed greater availability of water in those systems. Indeed, studied species improved their water use efficiency during the June–July–August period. This likely resulted in an increase in photosynthetic activity (A) in the studied tree species, then translated into higher productivity [99]. Furthermore, the shrubby conformation of *E. umbellata*, unlike Italian alder, implies lower evaporation of water thanks to a greater soil cover [78] ensuring large water availability and providing shelter and protection to the main species, particularly in the juvenile stages [100].

## 5. Conclusions

We found that *Populus alba* and *Juglans regia* increased their tree growth when growing with ancillary species, especially with N-fixing species, such as *A. cordata* and *E. umbellata*. Further, in all the intercropping systems, the mixture with hazel enhanced soil C stock.

*Populus alba* productivity increased especially in the juvenile phase (first 8 years), while in the adult phase we could speculate that resources competition among species became limiting (both aboveground and belowground), in particular with *C. avellana*.

Management practices, such as thinning and pruning, as well as scheduling a gradual removal of fast-growing species are thus crucial to favor the development of the principal species in these mixed plantations. Of the two intercropping systems with N-fixing species, the one with *E. umbellata* promoted water use efficiency better in both studied tree species, in particular in poplar.

Our study confirms our research hypothesis that companion species, especially with N-fixing species, can influence substantially wood formation and growth as well as soil C stock and highlights the importance of carefully considering the species dynamics in planning forest management.

**Supplementary Materials:** The following are available online at https://www.mdpi.com/article/10.3390/f12091251/s1, Figure S1: (A) Study area in which four intercropping systems were compared using a randomized blocks design with three replications. Photo modified by Google Earth, (B) Plant-

ing layout of the different intercropping. On the left poplar and walnut intercropping, on the right poplar and walnut intercropped respectively with hazel, italian alder and autumn-olive.

**Author Contributions:** G.B., F.P., G.C. and A.F. conceived and designed the study; F.N., T.D., M.I., G.M., G.B., M.C.M., F.P. and G.C. performed sampling; F.N. and G.B. performed dendrochronological and isotope analyses and data interpretation; T.D., M.I. and G.M. performed soil measurements; all authors contributed to interpretation of the overall data; F.N. and G.B. wrote the main part of the manuscript. All authors have read and agreed to the published version of the manuscript.

**Funding:** This research received no external funding.

**Institutional Review Board Statement:** Not applicable.

**Informed Consent Statement:** Not applicable.

**Data Availability Statement:** The data presented in this study are available on request from the corresponding author.

**Acknowledgments:** The author wishes to thank Vita Criscuolo for support in field activity and data curation. G. Battipaglia and F. Niccoli wish to thanks the VALERE PROJECT of University of Campania "Luigi Vanvitelli" and the MIUR-PRIN 2017 project "The Italian TREETALKER NETWORK: continuous large scale monitoring of tree functional traits and vulnerabilities to climate change".

**Conflicts of Interest:** The authors declare no conflict of interest.

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
