# Peer review of "Tree Species Composition in Mixed Plantations Influences Plant Growth, Intrinsic Water Use Efficiency and Soil Carbon Stock"

_forests, doi:10.3390/f12091251_

Round 1
Reviewer 1 Report
Thank you for the opportunity to study this paper.
The paper is well written and reports carefully designed research.
In conclusions, the authors state: “We found that at our study site Populus alba and Juglans regia increased their tree 467 growth and enhanced soil C stock when growing with ancillary species, especially with 468 N-fixing species, such as A. cordata and E. umbellata. This happened especially in the juve-469 nile phase (first 8 years), while in the adult phase we could speculate that resources com-470 petition among species became limiting (both aboveground and belowground), in partic-471 ular with C. avellana.”
It appears on the basis of Fig. 3 that the conclusion probably is correct in the case of Populus alba.
Fig. 5 appears to indicate that the situation possibly is the opposite in the case of Juglans regia: the difference mostly appears at mature age. Is that correct?
The paper requires linguistic revision.
The way of quoting references appears unusual.
Small remark: The reviewer feels it is a little superfluous to specify where the company distributing incremental borers resides.
Author Response
Q) Thank you for the opportunity to study this paper. The paper is well written and reports carefully designed research.
R) Dear Editor and Referee. We thank the reviewers for their careful reading of the manuscript and constructive remarks. We have taken the comments on board to improve and clarify the manuscript. In particular we have modified the sentences and paragraphs according reviewers’ suggestions. Further, a native English speaker has revised the paper. We believe that the manuscript is now more concise and clearer and we hope that the reviewers find it suitable for publication. The changes are highlighted in the text in red colour.
Please find below a detailed point-by-point response to all comments.
Looking forward to hearing from you
Giovanna Battipaglia
On behalf of all coauthors
Q1) In conclusions, the authors state: “We found that at our study site Populus alba and Juglans regia increased their tree 467 growth and enhanced soil C stock when growing with ancillary species, especially with 468 N-fixing species, such as A. cordata and E. umbellata. This happened especially in the juve-469 nile phase (first 8 years), while in the adult phase we could speculate that resources com-470 petition among species became limiting (both aboveground and belowground), in partic-471 ular with C. avellana.”
It appears on the basis of Fig. 3 that the conclusion probably is correct in the case of Populus alba.
Fig. 5 appears to indicate that the situation possibly is the opposite in the case of Juglans regia: the difference mostly appears at mature age. Is that correct?
R1) We modified the conclusion, following reviewer’s comments
Q2) The paper requires linguistic revision.
R2) The paper was reviewed by a native English speaker
Q3) The way of quoting references appears unusual.
R3) We reviewed the references, following journal guidelines
Q4) Small remark: The reviewer feels it is a little superfluous to specify where the company distributing incremental borers resides.
R4) We corrected it
Reviewer 2 Report
Very well structured article.
The conclusion is slightly outside the scope of the analysis and, in my opinion, it contains little speculation.
Table 3 is ambiguous (no enough description of letters: a - ANOVA b - post hoc c - ??) but intuitively easy to interpret.
Author Response
Dear Editor and Referee. We thank the reviewers for their careful reading of the manuscript and constructive remarks. We have taken the comments on board to improve and clarify the manuscript. In particular we have modified the sentences and paragraphs according reviewers’ suggestions. Further, a native English speaker has revised the paper. We believe that the manuscript is now more concise and clearer and we hope that the reviewers find it suitable for publication. The changes are highlighted in the text in red colour.
Please find below a detailed point-by-point response to all comments.
Looking forward to hearing from you
Giovanna Battipaglia
On behalf of all coauthors
Q1)Very well structured article.
R1)We thank the reviewer for her/his valuable comments.
Q2) The conclusion is slightly outside the scope of the analysis and, in my opinion, it contains little speculation.
R2) We modified the conclusion
Q3) Table 3 is ambiguous (no enough description of letters: a - ANOVA b - post hoc c - ??) but intuitively easy to interpret.
R3) We modified the table 3 writing “The superscript letters indicate significant differences between means in the different column and for each system, according to One-way ANOVA, followed by Student–Newman–Keuls coefficient for comparison tests (p < 0.05). Systems that do not share at least one letter are significantly different.”
Reviewer 3 Report
Review Niccoli et al 2021 Forests
Line 14 a word seem missing in the sentence “ efficiency of resources
Line 16 and elsewhere in the manuscript: consociate is a word I am not familiar with. I wonder if a more common word would not have conveyed the same meaning, like : companion species,
Line 18 consociation could be replaced by “ mixture”,
Line 25: seems a rather simplistic conclusion, should be detailed
Line 32 does this sentence apply worldwide or to a specific location ?
Line 34 article “the” is probably not necessary in the context. The authors should have their text reviewed by an English speaking reviewer.
Line 36 “increase of fire” is a fuzzy formulation
Line 37 and 38 seem simplistic and fuzzy.
Line 48 “flexibility to changes in the wood market” flexibility of what?
Line 54 “Guarantee” seems to strong of a word. Mixtures do not always reduce costs. Many times, they could increase costs.
Line 59 “in a meta-analysis”
Line 61 “endowed” does not seem to be the best word in this context. Why not simply “enriched”?
Line 69 check for occurrences of double spaces throughout the text ex. Line 465
Line 84 species mixtures
Line 93 insert an unbreakable space before “֯C”
Line 102 replace the hyphen in “715 trees ha1”
Line 148 measured by pre-treating…
Line 157 what do you mean by “shacked” platform?
Line 167 three-year rings
Line 182 and other equations and mathematical formulations are hard to read due to a lack spaces around mathematical operators. Check throughout the manuscript
Line 271 I don’t understand that sentence “the lowest growth right …”
Line 300, 304, 335 Shading in the tables is not necessary and should be removed. Also, too many lines are in the tables. Restrain to top and bottom of the table and below headings.
Line 358 to 369 would benefit from more recent references: check the following Loreau, M. & Hector, A. (2001). Partitioning selection and complementarity in biodiversity experiments. Nature, 412, 72-76.
Tobner CM, Paquette A, Reich PB, Gravel D, Messier C. 2014. Advancing biodiversity – ecosystem functioning science with the use of high-density tree-based experiments. Oecologia 174: 609-21. |
These mention the concept of “sampling effects” and “complementarity” effects which are similar to what the authors refer to as “principle of competitive production” and facilitation production principle.
Line 369. The mechanisms involved in facilitation are certainly not only by advantageous alteration of the local environment. Reading this I have a sense that you imply mostly the soil environment. Facilitation or complementarity could occur in the use of light (PRETZSCH, Hans. Canopy space filling and tree crown morphology in mixed-species stands compared with monocultures. Forest Ecology and Management, 2014, vol. 327, p. 251-264.) as well as in mediation of pests and diseases.
Line 377 nutrient availability
Line 386 the principal species
Line 387 not sure what you mean by “principal trees” is it trees of the principal species or something else.
Line 424 As for walnut…
Line 449 However, intercropping with …
Line 455 One the other hand, intercropping with …
Line 459 in this consociation. Indeed, studied species…
Line 473 check for the proper use of the word “scalar” what do you mean by it?
General comments: I wonder if the analysis could not include some form of overyielding calculations instead of analysing the production of species separately.
Author Response
Dear Editor and Referee
We thank the reviewers for their careful reading of the manuscript and constructive remarks. We have taken the comments on board to improve and clarify the manuscript. In particular we have modified the sentences and paragraphs according reviewers’ suggestions. Further, a native English speaker has revised the paper. We believe that the manuscript is now more concise and clearer and we hope that the reviewers find it suitable for publication.The changes are highlighted in the text in red colour.
Please find below a detailed point-by-point response to all comments.
Looking forward to hearing from you
Giovanna Battipaglia
On behalf of all coauthors
Q1) Line 14 a word seem missing in the sentence “ efficiency of resources
R1) We corrected the sentence
Q2) Line 16 and elsewhere in the manuscript: consociate is a word I am not familiar with. I wonder if a more common word would not have conveyed the same meaning, like : companion species,
R2) We have replaced "consociate" with "intercropping" , "companion species “ and “mixture”
Q3) Line 18 consociation could be replaced by “ mixture”,
R3) We used mixture
Q4) Line 25: seems a rather simplistic conclusion, should be detailed
R4) We modified the conclusion according to all reviewers’comments
Q5) Line 32 does this sentence apply worldwide or to a specific location ?
R5) We have added worldwide
Q6) Line 34 article “the” is probably not necessary in the context. The authors should have their text reviewed by an English speaking reviewer.
R6) We corrected it and the English has been revised
Q7) Line 36 “increase of fire” is a fuzzy formulation
R7) We have replaced it with “increase of fire frequency”
Q8) Line 37 and 38 seem simplistic and fuzzy.
R8) We rephrased the sentence. We added “To mitigate these negative effects and to safeguard the economic and ecological functions of forests, it is necessary to implement sustainable management of secondary forests”
Q9) Line 48 “flexibility to changes in the wood market” flexibility of what?
R9) We clarified the concept
Q10) Line 54 “Guarantee” seems to strong of a word. Mixtures do not always reduce costs. Many times, they could increase costs.
R10) We corrected it
Q11) Line 59 “in a meta-analysis”
R11) done
Q12) Line 61 “endowed” does not seem to be the best word in this context. Why not simply “enriched”?
R12) We corrected it
Q13) Line 69 check for occurrences of double spaces throughout the text ex. Line 465
R13) done
Q14) Line 84 species mixtures
R14) done
Q15) Line 93 insert an unbreakable space before “֯C”
R15) done
Q16) Line 102 replace the hyphen in “715 trees ha1”
R16) done
Q17) Line 148 measured by pre-treating…
R17) done
Q18) Line 157 what do you mean by “shacked” platform?
R18) We modified it, saying mechanical shaker
Q19) Line 167 three-year rings
R19) done
Q20) Line 182 and other equations and mathematical formulations are hard to read due to a lack spaces around mathematical operators. Check throughout the manuscript
R20) done
Q21) Line 271 I don’t understand that sentence “the lowest growth right …”
R21) We clarified the sentence.
Q22) Line 300, 304, 335 Shading in the tables is not necessary and should be removed. Also, too many lines are in the tables. Restrain to top and bottom of the table and below headings.
R22) done
Q23) Line 358 to 369 would benefit from more recent references: check the following Loreau, M. & Hector, A. (2001). Partitioning selection and complementarity in biodiversity experiments. Nature, 412, 72-76.
Tobner CM, Paquette A, Reich PB, Gravel D, Messier C. 2014. Advancing biodiversity – ecosystem functioning science with the use of high-density tree-based experiments. Oecologia 174: 609-21.
These mention the concept of “sampling effects” and “complementarity” effects which are similar to what the authors refer to as “principle of competitive production” and facilitation production principle.
R23) Thanks to the reviewer for this suggestion, we have added the references
Q24) Line 369. The mechanisms involved in facilitation are certainly not only by advantageous alteration of the local environment. Reading this I have a sense that you imply mostly the soil environment. Facilitation or complementarity could occur in the use of light (PRETZSCH, Hans. Canopy space filling and tree crown morphology in mixed-species stands compared with monocultures. Forest Ecology and Management, 2014, vol. 327, p. 251-264.) as well as in mediation of pests and diseases.
R24) We modified it in “through an advantageous alteration of the environmental conditions, such as greatest availability of light for the crown [65]”
Q25) Line 377 nutrient availability
R25) done
Q26) Line 386 the principal species
R26) done
Q27) Line 387 not sure what you mean by “principal trees” is it trees of the principal species or something else.
R27) We meant the valuable species Juglans regia and Populus alba.
Q28) Line 424 As for walnut…
R28) Done
Q29) Line 449 However, intercropping with …
R29) done
Q30) Line 455 One the other hand, intercropping with …
R30) done
Q31) Line 459 in this consociation. Indeed, studied species…
R31) done
Q32) Line 473 check for the proper use of the word “scalar” what do you mean by it?
R32) We clarified the sentence Adding gradual removal
Q33) General comments: I wonder if the analysis could not include some form of overyielding calculations instead of analysing the production of species separately.
R33) We thank the reviewer for this interesting suggestion. However, the evaluation of the total yield was not calculated as it did not represent decisive information for this research. Our study was interested in evaluating the productive yield of individual species to understand whether the different intercropping systems could or could not benefit their productivity.